# Pharmacological Therapies of Spinal Muscular Atrophy: A Narrative Review of Preclinical, Clinical–Experimental, and Real-World Evidence

**DOI:** 10.3390/brainsci13101446

**Published:** 2023-10-10

**Authors:** Salvatore Crisafulli, Brigida Boccanegra, Giacomo Vitturi, Gianluca Trifirò, Annamaria De Luca

**Affiliations:** 1Department of Medicine, University of Verona, Piazzale Ludovico Antonio Scuro 10, 37124 Verona, Italy; salvatore.crisafulli@univr.it; 2Department of Pharmacy-Drug Sciences, University of Bari “Aldo Moro”, Via E. Orabona 4, 70125 Bari, Italy; brigida.boccanegra@uniba.it (B.B.); annamaria.deluca@uniba.it (A.D.L.); 3Department of Diagnostics and Public Health, University of Verona, Piazzale Ludovico Antonio Scuro 10, 37124 Verona, Italy; giacomo.vitturi@univr.it

**Keywords:** spinal muscular atrophy, nusinersen, onasemnogene abeparvovec, risdiplam, pre-marketing evidence, post-marketing evidence

## Abstract

Spinal muscular atrophy (SMA) is a rare neuromuscular disease, with an estimated incidence of about 1 in 10,000 live births. To date, three orphan drugs have been approved for the treatment of SMA: nusinersen, onasemnogene abeparvovec, and risdiplam. The aim of this narrative review was to provide an overview of the pre- and post-marketing evidence on the pharmacological treatments approved for the treatment of SMA by identifying preclinical and clinical studies registered in clinicaltrials.gov and in the EU PAS register from their inception until the 4 January 2023. The preclinical evidence on the drugs approved for SMA allowed a significant acceleration in the experimental phase of these drugs. However, since these drugs had been authorized through accelerated programs, the conduction of post-marketing studies was requested as a condition of their marketing approval to better understand their risk–benefit profiles in real-world settings. As of the 4 January 2023, a total of 69 post-marketing studies concerning the three orphan drugs approved for SMA were identified in clinicaltrials.gov (N = 65; 94.2%) and in the EU PAS register (N = 4; 5.8%). Currently, ongoing studies are primarily aimed at providing evidence concerning the risk–benefit profile of the three drugs in specific populations that were not included in the pivotal trials and to investigate the long-term safety and clinical benefits of these drugs. Real-world data sources collecting information regarding the natural history of the disease and post-marketing surveillance of the available therapies are increasingly becoming essential for generating real-world evidence on this rare disease and its orphan drugs.

## 1. Introduction

Spinal muscular atrophy (SMA) is a rare, autosomal-recessive neuromuscular disease, with an estimated incidence of about 1 in 10,000 live births [1].

SMA is caused by biallelic mutations in the *SMN1* gene, encoding for the survival motor neuron (SMN) protein, that plays a key role in the proper functioning and survival of motor neurons. Insufficient levels of the SMN protein cause the progressive degeneration of motor neurons in the brainstem and spinal cord, leading to muscle weakness, motor difficulties, and atrophy of limbs, trunk, and respiratory muscles [2]. The impairment of the latter, in particular, is responsible for respiratory failure, which is the main cause of death in patients with SMA [3]. To date, genetic blood testing to identify depletions and mutations in the *SMN* gene is the first diagnostic step for patients with suspected SMA and represents the diagnostic reference standard [4].

SMA is classified into different phenotypes (i.e., SMA type 0, type 1, type 2, type 3, and type 4) based on the age of onset as well as the severity of the clinical conditions, which is inversely related to the amount of SMN protein available at the motor neuron level [1]. The phenotype–genotype correlation of the disease depends, in part, on the number of copies of a second gene, called *SMN2*, which is a paralogue of *SMN1*. In 90% of *SMN2* transcriptions, the *SMN2* transcript undergoes splicing of exon 7, thus yielding an unstable SMN∆7 protein, while, in the remaining 10%, a normal and functional SMN protein is produced, contributing to the survival of the spinal motor neurons and influencing the severity of the clinical manifestations [5].

To date, three orphan drugs have been approved for the treatment of SMA: nusinersen, an antisense oligonucleotide, onasemnogene abeparvovec, a gene therapy, and risdiplam, a *SMN2* pre-mRNA splicing small-drug modifier. Considering the seriousness of the disease and the urgent need for effective treatments, these drugs have been authorized through accelerated assessment procedures that reduce the timeframe for the European Medicines Agency (EMA) Committee for Medicinal Products for Human Use (CHMP) to review a marketing-authorization application from 210 to 150 days [6].

SMA is a clear example of how it is possible to implement strategies shared among different stakeholders (e.g., research centers, industries, and regulators) to build a solid path, based on robust preclinical, translational, and clinical evidence (in both pre- and post-marketing settings), aimed at supporting regulatory processes and ultimately providing patients with effective treatment strategies. This is particularly important for advanced therapies like onasemnogene abeparvovec, which have completely different pharmacokinetic and pharmacodynamic profiles than conventional therapies. 

The aim of this narrative review is to provide an overview of the pre- and post-marketing evidence on the pharmacological treatments approved for the treatment of SMA by identifying preclinical as well as clinical studies (both experimental and observational). With this aim, PubMed Central and Embase were searched using index terms (i.e., MeSH terms and emtree terms) related to SMA and its pharmacological therapies. Furthermore, relevant studies were also identified by searching two large repositories such as clinicaltrials.gov and the EU PAS register.

## 2. Pre-Marketing Evidence on SMA Pharmacological Therapies

### 2.1. Preclinical Studies

The deeper knowledge of the molecular mechanisms underlying SMA’s onset and progression has represented a significant milestone for the development of specific therapeutic strategies (Figure 1).

In this frame, animal testing provided a crucial help: several SMA animal models (*Caenorhabditis elegans*, zebrafish, and Drosophila) were indeed developed in the past. Nonetheless, only the setting up of a murine model able to accurately reproduce the human disease phenotype represented the actual cornerstone for preclinical studies [7]. Firstly, it must be considered that mice have only one SMN gene encoding for the SMN protein, instead of two as in humans. Moreover, homozygous mutations in the SMN gene induce early embryonic death in mice, whereas heterozygous SMN mice (SMN±) display a normal phenotype [8]. To overcome these issues, a novel transgenic murine model was developed, introducing the human *SMN2* gene in the SMN-null mice. This strategy made the murine disease phenotype more similar to the human one, also considering that *SMN2* manipulation allows for the introduction of specific patients’ mutations. This was also paralleled by the adoption of reliable and highly reproducible experimental protocols. The Translational Research in Europe for the Assessment and Treatment of Neuromuscular Diseases (TREAT-NMD) network [9] played a key role in this approach through the provision of internationally validated, standard operating procedures, useful to optimize protocols and accelerate the research process [10]. The timeline of the most important preclinical studies in the SMA field is summarized in Table 1.

#### 2.1.1. Nusinersen

Nusinersen was the first SMA orphan drug approved in 2016 by the Food and Drug Administration (FDA) and, in 2017, by the EMA. Nusinersen belongs to the class of antisense oligonucleotides (ASOs) and is able to target a specific splicing silencer site (ISS-N1, intronic splice silencing) located within the *SMN2* intron 7. As previously mentioned, small differences in the *SMN2* gene sequence led to alternative splicing processes, inducing the skipping of exon 7 in the *SMN2* transcript and the subsequent synthesis of a truncated, non-functional protein (SMNΔ7). Nusinersen prevents the binding of specific splicing repressors, hnRNPA1 and hnRNPA2, to ISS-N1, allowing the integration of exon 7 into the final transcript and the synthesis of a full-length SMN protein. Since 2006, systematic screenings have been conducted to find the most effective ASO. Several ASOs were tested in murine models and in patient-derived fibroblasts, shifting along the ISS-N1, one base at a time. Once the strongest sequence had been identified, the specific ASO was administered in SMA mice to evaluate the effects of the compound on in vivo and ex vivo disease readouts. These treatments markedly increased the SMN protein levels and ameliorated the disease phenotype, extending the median lifespan (severely reduced in the SMA mice) by 25-fold and leading to the significant recovery of motor functions (including α-motor neuron count, mean area of muscle fibers, heart weight, thickness of the interventricular septum, left ventricular wall, and integrity of neuromuscular junctions), which were all relevant effects from a clinical perspective [14]. Moreover, the pharmacokinetic analysis performed on murine models and non-human primates highlighted a good bioavailability of the compound at the central nervous system level if administered by intrathecal infusion. These promising data provided a solid base for the following successful clinical trials, ending with nusinersen’s approval for SMA treatment in children and adults. The main drawback of nusinersen use is represented by its inability to cross the blood–brain barrier (BBB) and the subsequent need for recurrent lumbar punctures for intrathecal administration. This considerably reduces therapeutic compliance, particularly for patients with severe scoliosis.

#### 2.1.2. Onasemnogene Abeparvovec

The replacement of the *SMN1* defective gene is an alternative therapeutic approach that has been recently explored. Onasemnogene abeparvovec was approved in 2020 as the first, and currently unique, gene therapy for SMA, and it was indicated for SMA patients aged <2 years with bi-allelic mutations in the *SMN1* gene and three or fewer copies of the *SMN2* gene or infantile-onset SMA. The small dimension of the *SMN1* gene facilitated its delivery via a recombinant self-complementary adeno-associated viral vector serotype 9 (scAAV9). Preclinical studies in mice and non-human primates demonstrated the ability of the drug to cross the BBB after intravenous injection. Furthermore, this gene therapy significantly improved lifespan (median of 282 days for treated mice vs. 17.5 days for untreated SMAΔ7 mice) and ameliorated motor functions, particularly in early-treated animals [19,20]. Toxicology studies confirmed the safety and good tolerability of the treatment after intravenous injection, opening for the translatability of the drug in humans using this specific route of administration. In addition, it must be underlined that preclinical studies were performed on juvenile animals: the very encouraging obtained results supported again the importance of early administration. In fact, intravenous injection in mice within 24 h of birth showed greater efficacy compared to a later administration (within 5 or 10 days of birth), thus further reinforcing the relevance of neonatal screenings for early treatment.

#### 2.1.3. Risdiplam

The success of nusinersen considerably boosted research for further therapies, in particular for those administered through a less invasive route. Innovative drug design techniques in combination with high-throughput screening methods permitted the identification of novel, small molecules that shared nusinersen’s mechanism of action, promoting the inclusion of exon 7. The first molecule of interest was an orally administered coumarin derivative, subsequently modified to overcome potential in vitro mutagenicity issues and the phototoxicity typically associated with this drug class. Further compound optimization ameliorated the selectivity and affinity to the target, allowing a reduction of the dose required to achieve the therapeutic goals while limiting the side effects. Finally, after these implementations, the pyrido-pyrimidinone derivative risdiplam was synthesized and successively approved by the FDA and EMA as the first oral drug for SMA treatment in 2020 and 2021, respectively [21]. Preclinical studies performed in mice and non-human primates confirmed the drug’s efficacy in restoring proper SMN protein levels and demonstrated a strong correlation between plasma and tissue concentration, overcoming the bioavailability issues of nusinersen [22]. In fact, several preclinical studies demonstrated that the SMN protein carries out a wide variety of actions at multiple levels and in different tissues, explaining the non-motor symptoms related to the pathology. Pharmacokinetic analyses documented the excellent distribution of risdiplam in several tissues, paving the way for an innovative therapy which combined an easier method of administration with the possibility to treat also peripheral symptoms. Real-world studies will help evaluate whether this systemic distribution could determine off-target effects, affecting the risk–benefit ratio of risdiplam.

### 2.2. Pivotal Clinical Trials

The characteristics of the pivotal clinical trials supporting the marketing approval of the three pharmacological therapies for the treatment of SMA are summarized in Table 2.

The efficacy of nusinersen was proved in the ENDEAR study, a 13-month double-blind phase 3 clinical trial, which enrolled 121 patients diagnosed with infantile-onset SMA before 6 months of age and who were under 7 months of age at the time of receiving the first dose [23]. The patients were randomized, in a 2:1 ratio, to receive an intrathecal injection of nusinersen (treatment group) or a sham procedure (control group). The primary endpoints included the achievement of a motor milestone response defined according to the Hammersmith Infant Neurological Examination and the event-free survival (i.e., time to death or requiring permanent assisted ventilation). The results of the per protocol analysis showed that 51% of the patients in the treatment group and no patients in the control group achieved a motor milestone response and that the likelihood of event-free survival was higher in the treatment group than in the control group (hazard ratio for death or the use of permanent assisted ventilation, 0.53; *p* = 0.005). However, the ENDEAR trial was terminated early due to the results of the interim analysis and to the ethical consideration for the patients in the control group, and this led to the loss of data and a short time period for the assessment of the safety and the efficacy of nusinersen.

Concerning onasemnogene abeparvovec, its efficacy was demonstrated in the STR1VE-US study, an open-label phase 3 clinical trial including 22 symptomatic patients diagnosed with SMA type 1 who were younger than 6 months [24]. The enrolled patients were followed-up until the age of 18 months. The co-primary efficacy endpoints were independent sitting for ≥30 s at the study visit at 18 months of age and event-free (i.e., time to death or requiring permanent ventilation) survival at 14 months of age. Such endpoints were compared to 23 untreated infants aged ≤6 months with SMA type 1 from the Pediatric Neuromuscular Clinical Research (PNCR) dataset, a historical cohort of untreated infants with SMA type 1 [27]. The first co-primary endpoint was met by 13 (59%) of the 22 patients in the treatment group vs. 0 patients in the PNCR cohort, while the second was achieved by 20 (91%) patients in the treatment group vs. 6 (26%) in the PNCR cohort. The main limitations of the STR1VE-US trial were the lack of a randomized comparison to a control group, potentially leading to an overestimation of the treatment effect, the lack of masking for both intervention and outcome evaluation, the strict inclusion criteria, and the short follow-up period.

The marketing approval of risdiplam was based on two pivotal clinical trials: the FIREFISH trial in infantile-onset SMA [26] and the SUNFISH trial in late-onset SMA [25]. The FIREFISH study was an open-label, single-arm study including 41 infants aged between 1 and 7 months, with genetically confirmed Type 1 SMA. The primary endpoint was the ability to sit without support for at least 5 s after 12 months of treatment, a milestone that is never achieved in untreated patients affected by Type 1 SMA. After 12 months of treatment, 12 patients (29%) met this endpoint [26]. The internal validity of the FIREFISH trial was affected by the single-arm design, precluding a precise estimation of the magnitude of the benefit.

The SUNFISH study was a phase 3, randomized, double-blind, placebo-controlled clinical trial in patients aged between 2 and 25 years with confirmed type 2 or type 3 SMA. Patients were randomized, in a 2:1 ratio, to receive daily oral risdiplam (treatment group, N = 120) or daily oral placebo (control group, N = 60), and they were followed-up for 12 months. The primary endpoint was the change from the baseline in the 32-item Motor Function Measure (MFM32)’s total score at month 12. After one year, the least squares mean change from the baseline was 1.36 (95% confidence interval 0.61 to 2.11) in the risdiplam group and −0.19 (−1.22 to 0.84) in the placebo group, with a treatment difference of 1.55 (0.30 to 2.81, *p* = 0.016) in favor of risdiplam. However, the improvement in motor function was mainly detected in the younger patients (i.e., children aged between 2 and 5 years) rather than in the adolescent and adult patients, thus leaving unanswered questions for risdiplam use in adults with SMA when considering efficacy and long-term expectations [28].

## 3. Post-Marketing Evidence

Post-marketing studies, including both observational studies and clinical trials, play a crucial role in better understanding the clinical benefits and the safety of orphan drugs in real-world settings [29]. This holds particularly true for orphan drugs that are usually marketed through conditional approval and other accelerated programs based on the availability of limited pre-marketing evidence on their benefit–risk profile. As such, the conduction of further clinical trials and even more observational studies to generate post-marketing evidence has been requested as part of the risk-management plans of nusinersen [30], onasemnogene abeparvovec [31], and risdiplam [32], as a condition of their marketing approval (Table 3).

Planned, ongoing, and finalized clinical trials, as well as observational studies, concerning the three orphan drugs approved for the treatment of SMA were searched in two study repositories: clinicaltrials.gov and the European post-authorization study (EU PAS) Register from their inception until the 4 January 2023.

Clinicaltrials.gov is a public repository created in 1997 and run by the United States National Library of Medicine at the National Institutes of Health, in which interventional and observational studies receiving US government funding, as well as studies funded by commercial entities and programs providing access to investigational drugs outside of clinical trials (i.e., expanded access), are registered. It was searched for the following keywords: “spinal muscular atrophy” and “nusinersen” or “onasemnogene abeparvovec” or “risdiplam”.

The EU PAS register is a publicly available register of non-interventional post-authorisation studies (PASS) that was launched in 2010. It is established and maintained by the EMA through the European Network of Centres for Pharmacoepidemiology and Pharmacovigilance (ENCePP), with the aims of increasing transparency, promoting the exchange of information, and ensuring the compliance with the European pharmacovigilance legislation requirements [33]. The current European pharmacovigilance legislation requires the public availability of study protocols and summary of results for PASS to be imposed as an obligation of marketing authorization by competent authorities. This register is also aimed at hosting non-imposed studies, such as those required as per the risk management plan, and all observational studies performed on authorized medicinal products, including effectiveness studies.

For each study, we extracted data on study design (i.e., interventional, observational, expanded-access study), study drug, status, primary purpose, study phase, and patients’ age.

As of the 4 January 2023, a total of 69 studies concerning the three orphan drugs approved for SMA were identified in clinicaltrials.gov (N = 65; 94.2%) and in the EU PAS register (N = 4; 5.8%) (Table 3). Thirty-nine (56.5%) of them were interventional, twenty-seven (39.1%) observational, and three (4.4%) were expanded-access programs. The majority of these studies investigated nusinersen (N = 34; 49.3%), followed by onasemnogene abeparvovec (N = 15; 21.7%), and risdiplam (N = 15; 21.7%); one study concerned both nusinersen and risdiplam, and one study concerned all three drugs. Concerning the status of the studies, forty-four (62.0%) studies were ongoing or planned, nineteen (27.5%) were completed, two (2.9%) were terminated or withdrawn, while, for four studies, the status was not reported (5.8%). As for the funder type, 52 (75.4%) of the 69 studies were funded by pharmaceutical companies, while 17 (24.6%) were funded by universities or research organizations.

The ongoing studies were mainly aimed at providing further evidence concerning the effectiveness of the three drugs on disease progression in specific populations that had not been included in the pivotal trials (e.g., adults, older adults, and pregnant women) and at investigating the long-term safety and clinical benefits of these drugs.

The characteristics of the 69 studies identified in the two repositories are reported in Table 4.

The median size of the enrolled populations in the identified interventional studies was 33 (interquartile range: 24–130.5). Most of these studies (N = 30, 76.9%) were open-label clinical trials and, mostly, single-group trials. Concerning the observational studies, half of them (N = 14; 51.8%) were prospective studies, and the median number of enrolled patients was 50.5 (interquartile range: 26.5–195).

The studies included in the risk-management plan of nusinersen were the following: two open-label extension studies (i.e., the SHINE and the NURTURE trials) investigating the long-term safety, tolerability, and efficacy of repeated doses of nusinersen administered through intrathecal injections in infants with genetically diagnosed and pre-symptomatic SMA; two registry-based observational studies (i.e., the TREAT-NMD network [9] and the Muscular Dystrophy Association United States (MDA US) Neuromuscular Disease Registry [34]); and one natural history study (i.e., the ISMAC study) [35]. 

The studies included in the risk-management plan of onasemnogene abeparvovec were three phase 3 clinical trials (i.e., the STR1VE-US, the SPR1NT, and the STR1VE-US trials) and three observational studies, including a prospective observational registry (i.e., the RESTORE study) and two long-term follow-up studies (i.e., the START and the AVXS-101-LT-002 studies) aimed at collecting safety data of patients with SMA treated using this drug.

Finally, the risk-management plan of risdiplam included four open-label extension studies (i.e., the FIREFISH, the SUNFISH, the JEWELFISH, and the RAINBOWFISH trials) and two observational studies aimed at collecting data concerning selected pregnancy outcomes and complications in women with SMA receiving risdiplam (Study BN42833) and at estimating the effects of single, oral doses of risdiplam on the QT interval of the electrocardiogram (Study BP42817).

Real-world data (RWD), defined as data relating to a patient’s health status and the delivery of health services that are collected during daily clinical practice, have a considerable potential for the evaluation of the safety and effectiveness of orphan drugs, as well as for providing relevant information concerning the epidemiology and the natural history of rare diseases. Real-world evidence (RWE), i.e., the evidence generated through the analysis of RWDs, therefore, plays a crucial role in better understanding the benefit–risk profile of orphan drugs and in supporting regulatory processes. Indeed, regulatory agencies are increasingly promoting the use of RWD for supporting regulatory processes, especially in the field of rare diseases. In particular, the EMA and the FDA promote the use of RWD for supporting drug approval, especially when the latter is based on limited pre-marketing evidence [36,37].

Disease registries provide very useful data not only for studying the clinical course of diseases and for conducting natural history studies, but also for identifying prognostic factors and for evaluating the clinical outcomes of available therapies [38]. In particular, the ISMAC natural history study is an ongoing prospective cohort study conducted by three national networks in the United States, Italy, and the United Kingdom, aimed at recording information that allows researchers to phenotype SMA patients and at following them over the years [39]. The TREAT-NMD is a registry collecting data from natural history studies conducted in several countries with the aim of gaining more information on the natural history of SMA, thus providing a context to better understand the safety and efficacy of new treatments and support their post-marketing surveillance [9]. Finally, the MDA US Neuromuscular Disease Registry is a prospective longitudinal registry aimed at collecting data on four neuromuscular diseases (i.e., SMA, amyotrophic lateral sclerosis, and Duchenne and Becker muscular dystrophies), with the purpose of facilitating translational research to improve standards of care and patients’ clinical outcomes [34].

## 4. Current Knowledge about the Risk–Benefit Profile of the Drugs Approved for the Treatment of SMA

To date, several studies summarizing mid- and long-term follow-up in SMA patients treated with nusinersen, risdiplam, and onasemnogene abeparvovec have been published. Such studies provide interesting data about the effectiveness and the safety of these innovative therapies. As expected, the largest amount of data concerns nusinersen, since it received the FDA’s approval first. A prospective 3-year registry study (SMArtCARE), published in January 2023, highlighted a stabilization of the disease’s progression in 231 ambulant patients (114 pediatric patients with a median age of 8.6 years and 117 adult patients with a median age of 37 years), in parallel with an improvement in motor function [40]. In total, 31 of the pediatric patients (27.2%) and 31 of the adult patients (26.5%) achieved an improvement of more than 30 m in the 6-minutes WalkTest. On the contrary, among the patients receiving nusinersen, only five adult patients (7.7%) experienced a decline in walking a distance over 30 m, and two pediatric patients (1.8%) lost the ability to walk without assistance [40]. Another observational study published in March 2023 reported the results of a 4-year follow-up in 48 patients aged between 7 days and 12 years who had been treated with nusinersen [41]. The study confirmed the safety and the efficacy of the therapy over time, with evidence of a mild but significant amelioration of disease-related neuromuscular symptoms evaluated through the Children’s Hospital of Philadelphia Infant Test of Neuromuscular Disorders (CHOP INTEND) and the Hammersmith Infant Neurological Examination (HINE-II). During the 4-year follow-up period, only one patient receiving nusinersen died between the second and the fourth year of follow-up [41].

More recently, Novartis rolled out relevant data regarding a long-term follow-up in patients treated with onasemnogene abeparvovec [42]. A single intravenous infusion administered to symptomatic SMA patients ensured the maintenance of the motor milestones previously achieved, even after 7.5 years. Importantly, preliminary results of the LT-002 study showed that all of the 25 enrolled children, treated during the pre-symptomatic phase, maintained the highest milestone previously achieved during the parent study, according to the Developmental Milestone Checklist, and independent walking. These results underline the importance of early interventions to minimize motor-function loss, according to the accepted “time-is-neuron” view [43].

On February 2023, after two fatal cases of acute liver failure were reported in SMA patients treated with onasemnogene abeparvovec, the EMA Pharmacovigilance Risk Assessment Committee (PRAC) disseminated a Direct Healthcare Professional Communication (DHPC) to alert physicians about the risk of hepatotoxicity [44].

Overall, the evidence on the pharmacological therapies for SMA is still limited and should be cautiously interpreted. The majority of the published studies analyzed involved only children and mainly focused on the motor outcomes, while only few of them investigated the effects on the bulbar and respiratory items. As such, several questions regarding the benefit–risk profile of these treatments are still unanswered, especially regarding the long-term effects of these therapies on patients’ survival and quality of life. To date, no studies regarding the mid- or long-term safety and effectiveness on risdiplam have been published, although a recruitment phase for a long-term follow-up study is currently ongoing [45].

In the next years, real-world studies will play a key role in providing evidence on the long-term safety as well as the disease’s stabilization in both pediatric and adult patients treated with the three treatments currently approved for the treatment of SMA. Furthermore, real-world experiences will be useful to guide patients and caregivers in the choice of the most suitable therapeutic option or, eventually, for changing therapy if a lack of efficacy is observed.

Despite the considerable advances in scientific research concerning the pharmacological treatments of SMA, some issues related to each of the three approved drugs remain, and caution is needed when interpreting study results, especially for adult patients. As for nusinersen, in addition to the above-mentioned administration issues, the lack of efficacy in controlling peripheral symptoms should also be considered. In fact, the SMN protein is also located in extra central nervous system organs (such as muscle, heart, pancreas, afferent nerves, bones, and gastroenteric system) and low levels of SMN induce a wide variety of non-motor symptoms (i.e., impaired skeletal muscle development, cardiomyopathy, altered liver function, severe diarrhea, and lower bone density) [46,47]. Regarding onasemnogene abeparvovec, the long-term effects of gene therapy are not known yet, and the therapeutic effect produced by this drug is not as valid in patients with late-onset SMA. Finally, as for risdiplam, the systemic effect followed by its administration could produce delayed side effects caused by a large-scale synthesis of the SMN protein.

## 5. Conclusions

Preclinical and translational studies are currently being carried out to discover and test potentially new therapies for the treatment of SMA, including the use of innovative platforms such as organoids. Furthermore, several interventional, as well as observational, studies focusing on the three available orphan drugs approved for SMA treatment are currently ongoing, with the aim of generating post-marketing evidence to better understand their risk–benefit profile.

Researchers are also putting great efforts in enhancing therapeutic strategies with innovative or repositioned drugs that can influence neuronal death processes or peripheral consequences due to the lack of the SMN protein. In this regard, translational research acquires an increasingly important role in accelerating drug development, and real-world data sources such as disease registers, collecting information regarding the natural history of the disease and post-marketing surveillance of the available therapies, are increasingly becoming essential tools to generate real-world evidence on rare disease with the dual aim of refining preclinical research and supporting the regulatory processes of orphan drugs.

## Figures and Tables

**Figure 1 brainsci-13-01446-f001:**
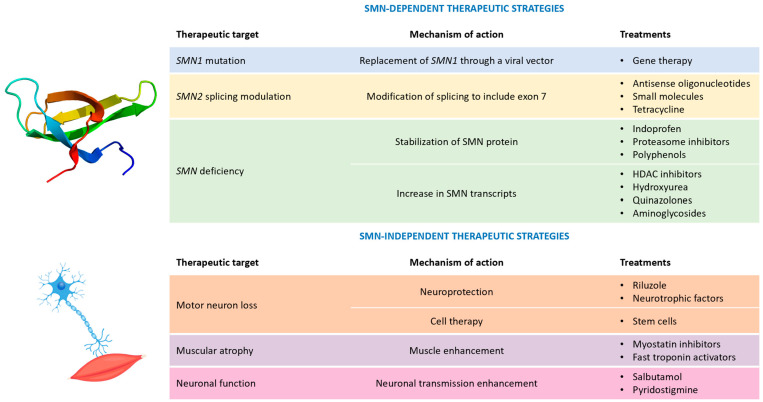
Possible therapeutic strategies for spinal muscular atrophy. Legend: HDAC = histone deacetylase; and SMN = survival motor neuron.

**Table 1 brainsci-13-01446-t001:** Timeline of preclinical studies of orphan drugs approved for spinal muscular atrophy treatment.

Drug	Mechanism of Action	Timeline	Detected Issues
Nusinersen	Inclusion of exon 7 in *SMN2* transcript	2006: Identification of ISS-N1 sequence located within *SMN2* intron 7 and synthesis of the first complementary ASO [11] 2008: Synthesis of ASO 10-27 with high affinity to ISS-N1; first preclinical studies on SMA animal models (rodents) [12] 2010–2011: The administration of ASO 10-27 via intrathecal or intracerebroventricular injection in SMA mice improved the expression levels of the SMN protein and ameliorated the disease phenotype; no increase in mice lifespan was detected [13] 2011: Subcutaneous injection of ASO 10-27 in SMA mice induced an amelioration of peripheral symptoms and improved lifespan by more than 25 times [14]. Intrathecal injection in non-human primates ensured a proper distribution at spinal cord level, without significant side effects [13]	Not able to cross the blood–brain barrier. Administered only via intrathecal injection.
Onasemnogene abeparvovec	Replacement of *SMN1* gene	2009–2010: construction of a recombinant adeno-associated viral vector serotype 9 (scAAV9) including the *SMN1* gene [15] 2010–2013: several preclinical studies conducted on the SMA murine model revealed that the drug had a high efficiency in improving lifespan and reducing the severity of symptoms [16]	Uncertainties regarding treatment response and long-term outcomes. Late administration not useful.
Risdiplam	Inclusion of exon 7 in *SMN2* transcript	2014: An orally administered coumarin derivative with a similar mechanism of action to nusinersen’s was identified through drug design techniques in combination with high-throughput screening methods. In vitro test on patient-derived fibroblasts showed an increase in SMA protein levels and the oral administration in SMA mice improved motor function and lifespan [17] 2016: Drug optimization techniques led to the synthesis of the compound RG7800 which passed toxicity and mutagenicity tests. The molecule did not exhibit phototoxicity issues. The administration in SMA mice confirmed the encouraging results obtained with the lead compound. Nonetheless, retinal toxicity was detected after the administration in non-human primates [18] 2018: Further modifications of the RG7800 structure led to the synthesis of risdiplam, which displayed a higher affinity to the target and an excellent systemic bioavailability. The drug resulted safe and well-tolerated after oral administration in SMA animal models	The systemic action could induce yet unknown side effects.

Abbreviations: ASO = antisense oligonucleotide; SMA = spinal muscular atrophy; and SMN = survival motor neuron.

**Table 2 brainsci-13-01446-t002:** Pivotal clinical trials leading to the marketing authorization of nusinersen, onasemnogene abeparvovec, and risdiplam.

Study	Study Design	Status	Eligibility Criteria	Exposure		Outcome	Results
				Treatment Group	Control Group		
Nusinersen							
ENDEAR [23]	Phase III double-blind, controlled, randomized clinical trial.	Completed	Patients with homozygous deletions or mutations in the *SMN1* gene and with two copies of the *SMN2* gene and younger than 7 months at the screening date	80 patients treated with nusinersen at a dose adjusted by cerebrospinal fluid volume, such that the patients 2 years old or older received a dose equivalent to 12 mg.	41 patients treated with a sham procedure	Achievement of a motor milestone (defined based on the Hammersmith Infant Neurological Examination results) Event-free survival (i.e., death or lifelong use of assisted ventilation)	51% (37/73) of the patients in the treatment arm achieved at least one motor milestone, compared with 0 out of the 37 patients in the control arm. 61% (49/80) of the patients did not die or did not require permanent ventilation (compared to 32% [13/41] of the patients in the control group; *p* = 0.005) at the end-of-trial visit (i.e., day 183, 302, or 394).
Onasemnogene abeparvovec							
STR1VE-US [24]	Phase III open-label, single-arm clinical trial	Completed	Patients with SMA type 1, with biallelic *SMN1* mutations, with one or two copies of *SMN2*, and younger than 6 months of age at the date of onasemnogene abeparvovec administration	22 patients treated with a single intravenous administration of onasemnogene abeparvovec, over approximately 60 min: 1.1 × 10^14^ vg/kg	23 untreated children, aged less than 6 months and with SMA type 1 registered in the Pediatric Neuromuscular Clinical Research dataset (historical controls)	Proportion of patients able to sit unassisted for at least 30 s at the 18 months follow-up visit Survival to the age of 14 months (i.e., no death or permanent ventilation)	13 out of the 22 patients (59%) were able to sit unassisted for 30 s or more at 18 months of age (compared with 0 out of the 23 untreated patients in the Pediatric Neuromuscular Clinical Research cohort; *p* < 0.0001). 20 out of the 22 patients (91%) survived without a need for permanent ventilation use for up to 14 months (vs. 6/23 [26.1%] of the untreated patients in the Pediatric Neuromuscular Clinical Research cohort; *p* < 0.0001).
Risdiplam							
SUNFISH—Part 2 [25]	Phase III double-blind, placebo-controlled, randomized clinical trial	Ongoing	Patients between 2 and 25 years of age, with type 2 or type 3 SMA	120 patients treated with oral-administered risdiplam (5 mg/day for patients weighing ≥ 20 kg or 0.25 mg/kg/day for patients weighing < 20 kg)	60 placebo-treated patients	Variation from the baseline in the 32-item Motor Function Measure’s total score for patients at the 12th month of treatment compared to the placebo group	At month 12, the mean change from baseline in the MFM32 score was 1.36 (95% CI: 0.61 to 2.11) in the treatment group and −0.19 (95% CI: −1.22 to 0.84), with a difference of 1.55 (95% CI: 0.30–2.81; *p* = 0.016) in favor of the treatment with risdiplam.
FIREFISH—Part 2 [26]	Phase II/III open-label, single-arm clinical trial	Completed	Patients aged 1 to 7 months with a type 1 SMA diagnosis and two copies of the *SMN2* gene	41 patients treated with risdiplam. The patients older than 5 months received a dose of 0.2 mg/kg/day; the patients younger than 5 months received a dose of 0.04 or 0.008 mg/kg/day, adjusted to 0.2 mg/kg/day within 1–3 months from starting treatment.	16 historical controls derived from the NeuroNEXT study, and 24 historical controls	Proportion of patients able to sit without assistance for at least 5 s after 12 months of treatment, based on the 3rd edition of the Bayley Scales of Infant and Toddler Development	12 out of the 41 patients (29%; 95% CI: 16 to 46) were able to sit without assistance for at least 5 s after 12 months of treatment; the percentage was significantly higher than in the historical controls (*p* < 0.001).

Abbreviations: CI = confidence interval; MFM32 = 32-item Motor Function Measure; SMA = spinal muscular atrophy; SMN = spinal motor neuron; and vg = vector genomes.

**Table 3 brainsci-13-01446-t003:** Characteristics of the post-marketing clinical studies concerning marketed orphan drugs that are approved for SMA, as identified in clinicaltrials.gov and the EU PAS register on the 4 January 2023.

	Studies Registered in Clinicaltrials.gov N = 65 (%)	Studies Registered in the EU PAS Register N = 4 (%)	Total N = 69 (%)
Study design			
Interventional	39 (60.0)	0 (0.0)	39 (56.5)
Observational	23 (35.4)	4 (100)	27 (39.1)
Expanded-access study	3 (4.6)	0 (0.0)	3 (4.4)
Study drug			
Nusinersen	33 (50.8)	1 (25.0)	34 (49.3)
Onasemnogene abeparvovec	14 (21.5)	1 (25.0)	15 (21.7)
Risdiplam	13 (20.0)	2 (50.0)	15 (21.7)
More than one	2 (3.1)	0 (0.0)	2 (2.9)
Not reported	3 (4.6)	0 (0.0)	3 (4.4)
Status of the study			
Recruiting	22 (33.8)	N.A.	22 (31.9)
Completed	19 (29.2)	N.A.	19 (27.5)
Active, not recruiting	12 (18.5)	N.A.	12 (17.4)
Not yet recruiting	6 (9.2)	N.A.	6 (8.7)
Unknown status	4 (6.2)	N.A.	4 (5.8)
Terminated	1 (1.5)	N.A.	1 (1.4)
Withdrawn	1 (1.5)	N.A.	1 (1.4)
Ongoing	N.A.	2 (50.0)	2 (2.9)
Planned	N.A.	2 (50.0)	2 (2.9)
Primary purpose			
Treatment	35 (53.8)	N.A.	35 (50.7)
Effectiveness evaluation	N.A.	2 (50.0)	2 (2.9)
Safety evaluation	N.A.	1 (25.0)	1 (1.4)
Other *	27 (41.5)	1 (25.0)	28 (40.6)
Not specified	3 (4.6)	0 (0.0)	3 (4.4)
Study phase			
Phase 1	10 (15.4)	N.A.	10 (14.5)
Phase 1 and 2	1 (1.5)	N.A.	1 (1.4)
Phase 2	5 (7.7)	N.A.	5 (7.3)
Phase 2 and 3	4 (6.2)	N.A.	4 (5.8)
Phase 3	12 (18.5)	N.A.	12 (17.4)
Phase 4	4 (6.2)	N.A.	4 (5.8)
Not applicable	23 (35.4)	4 (100)	27 (39.1)
Not reported	6 (9.2)	N.A.	6 (8.7)
Patients’ age			
Child	21 (32.3)	0 (0.0)	21 (30.4)
Child, Adult	10 (15.4)	2 (50.0)	12 (17.4)
Child, Adult, Older adult	21 (32.3)	2 (50.0)	23 (33.3)
Adult	5 (7.7)	0 (0.0)	5 (7.3)
Adult, Older adult	8 (12.3)	0 (0.0)	8 (11.6)

Abbreviations: N.A. = not applicable; SMA = spinal muscular atrophy; and * Other = basic science, diagnostic, supportive care, prevention.

**Table 4 brainsci-13-01446-t004:** Description of the post-marketing clinical studies of the orphan drugs marketed for SMA treatment as identified in clinicaltrials.gov and in the EU PAS register on the 4 January 2023.

NCT or EU PAS Number	Status	Study Drug	Age	Phase	Study Size	Study Type	Study Design	Primary Purpose	Funded by	Country	Included in the Risk Management Plan
clinicaltrials.gov											
NCT01494701	Completed	Nusinersen	Child	1	28	Interventional	Non-randomized, open-label clinical trial	Treatment	Industry	United States	No
NCT01703988	Completed	Nusinersen	Child	1/2	34	Interventional	Non-randomized, open-label clinical trial	Treatment	Industry	United States	No
NCT01780246	Completed	Nusinersen	Child	1	18	Interventional	Single-group, open-label clinical trial	Treatment	Industry	United States	No
NCT01839656	Completed	Nusinersen	Child	2	21	Interventional	Non-randomized, open-label clinical trial	Treatment	Industry	United States, Canada	No
NCT02052791	Completed	Nusinersen	Child, Adult, Older Adult	1	47	Interventional	Single-group, open-label clinical trial	Treatment	Industry	United States	No
NCT02292537	Completed	Nusinersen	Child	3	126	Interventional	Quadruple-blinded, randomized clinical trial	Treatment	Industry	United States, Canada, France, Germany, Hong Kong, Italy, Japan, Republic of Korea, Spain, and Sweden	No
NCT02386553	Active, not recruiting	Nusinersen	Child	2	25	Interventional	Single-group, open-label clinical trial	N.R.	Industry	United States, Australia, Germany, Italy, Qatar, Taiwan, and Turkey	Yes
NCT02462759	Terminated	Nusinersen	Child, Adult, Older Adult	2	21	Interventional	Quadruple-blinded, randomized clinical trial	Treatment	Industry	United States, Germany	No
NCT02594124	Active, not recruiting	Nusinersen	Child, Adult, Older Adult	3	292	Interventional	Triple-blinded, non-randomized clinical trial	Treatment	Industry	United States, Australia, Canada, France, Germany, Hong Kong, Italy, Japan, Spain, Sweden, Turkey, and United Kingdom	Yes
NCT04050852	Withdrawn	Nusinersen	Child, Adult	1	0	Interventional	Single-group, open-label clinical trial	Treatment	University	United States	No
NCT04089566	Recruiting	Nusinersen	Child, Adult, Older Adult	2/3	145	Interventional	Double-blinded, randomized clinical trial	Treatment	Industry	United States, Brazil, Canada, Chile, Colombia, Estonia, France, Germany, Greece, Hungary, Ireland, Israel, Italy, Japan, and Republic of Korea	No
NCT04159987	Recruiting	Nusinersen	Adult, Older Adult	N.A.	20	Interventional	Single-group, open-label clinical trial	Treatment	University	France	No
NCT04488133	Recruiting	Nusinersen	Child	4	60	Interventional	Single-group, open-label clinical trial	Treatment	Industry	United States, Germany, Italy, Japan, and Spain	No
NCT04576494	Recruiting	Nusinersen	Adult, Older Adult	N.A.	24	Interventional	Single-group, open-label clinical trial	Treatment	University	France	No
NCT04674618	Recruiting	Nusinersen	Adult, Older Adult	N.A.	58	Interventional	Open-label, randomized clinical trial	Treatment	University	Italy	No
NCT04729907	Recruiting	Nusinersen	Child, Adult, Older Adult	3	172	Interventional	Quadruple-blinded, non-randomized clinical trial	Treatment	Industry	United States, Brazil, Canada, Chile, Colombia, Estonia, Germany, Hungary, Japan, Mexico, Russia, Saudi Arabia, Spain, and Taiwan	No
NCT05067790	Recruiting	Nusinersen	Child, Adult	3	135	Interventional	Single-group, open-label clinical trial	Treatment	Industry	United States, Belgium, Germany, Italy, Poland, and Spain	No
NCT02865109	Unknown status	Nusinersen	Child, Adult, Older Adult	N.A.	N.A.	Expanded Access	N.R.	Treatment	Industry	Colombia, New Zealand, and Turkey	No
NCT03709784	Active, not recruiting	Nusinersen	Adult, Older Adult	N.A.	48	Observational	Prospective cohort study	N.R.	University	United States, Canada	No
NCT03878030	Active, not recruiting	Nusinersen	Adult	N.A.	12	Observational	Prospective cohort study	N.R.	Community-based organization	United States	No
NCT04587492	Completed	Nusinersen	Child, Adult	N.A.	35	Observational	Prospective cohort study	N.R.	University	Slovenia	No
NCT04591678	Completed	Nusinersen	Adult	N.A.	15	Observational	Prospective cohort study	N.R.	Industry	United States	No
NCT04602195	Recruiting	Nusinersen	Child	N.A.	60	Observational	Prospective observational study	N.R.	University	France	No
NCT04644393	Unknown status	Nusinersen	Child	N.A.	40	Observational	Retrospective observational study	N.R.	University	France	No
NCT04419233	Active, not recruiting	Nusinersen	Child, Adult, Older Adult	N.A.	50	Observational	Case-only study	N.R.	Industry	China	No
NCT04317794	Recruiting	Nusinersen	Child, Adult, Older Adult	N.A.	145	Observational	Case-only study	N.R.	Industry	Republic of Korea	No
NCT04404764	Completed	Nusinersen	Child, Adult, Older Adult	N.A.	155	Observational	Cross-sectional study	N.R.	University	Brazil	No
NCT05187260	Recruiting	Nusinersen	Child, Adult	N.A.	1000	Observational	Prospective cohort study	N.R.	University	China	No
NCT05644899	Not yet recruiting	Nusinersen	Adult, Older Adult	N.A.	51	Observational	Retrospective cohort study	N.R.	University	Italy	No
NCT04825119	Recruiting	Nusinersen	Child, Adult, Older Adult	N.A.	110	Observational	Prospective cohort study	N.R.	University	Slovenia	No
NCT04139343	Recruiting	Nusinersen	Child, Adult, Older Adult	N.A.	140	Observational	Prospective case–control study	N.R.	University	United States	No
NCT05042921	Not yet recruiting	Nusinersen	Child, Adult	N.A.	300	Observational	Prospective cohort study	N.R.	Industry	N.R.	No
NCT05291962	Completed	Nusinersen	Child, Adult	N.A.	14	Observational	Retrospective observational study	N.R.	University	Turkey	No
NCT02122952	Completed	Onasemnogene abeparvovec	Child	1	15	Interventional	Single-group, open-label clinical trial	Treatment	Industry	United States	No
NCT03381729	Completed	Onasemnogene abeparvovec	Child	1	32	Interventional	Non-randomized, open-label clinical trial	Treatment	Industry	United States	No
NCT03461289	Completed	Onasemnogene abeparvovec	Child	3	33	Interventional	Single-group, open-label clinical trial	Treatment	Industry	Belgium, France, Italy, and United Kingdom	No
NCT03505099	Completed	Onasemnogene abeparvovec	Child	3	30	Interventional	Single-group, open-label clinical trial	Treatment	Industry	United States, Australia, Belgium, Canada, Japan, and United Kingdom	No
NCT03837184	Completed	Onasemnogene abeparvovec	Child	3	2	Interventional	Single-group, open-label clinical trial	Treatment	Industry	Japan, Republic of Korea, and Taiwan	No
NCT04042025	Active, not recruiting	Onasemnogene abeparvovec	Child, Adult, Older Adult	3	85	Interventional	Single-group, open-label clinical trial	Treatment	Industry	United States, Australia, Belgium, Canada France, Italy, Japan, United Kingdom	Yes
NCT04851873	Recruiting	Onasemnogene abeparvovec	Child	3	24	Interventional	Single-group, open-label clinical trial	Treatment	Industry	United States, Australia, Belgium, Canada, France, Italy, Portugal, Switzerland, Taiwan, United Kingdom	No
NCT05073133	Active, not recruiting	Onasemnogene abeparvovec	Child	4	16	Interventional	Single-group, open-label clinical trial	Treatment	Industry	Argentina, Brazil	No
NCT05089656	Recruiting	Onasemnogene abeparvovec	Child	3	125	Interventional	Quadruple-blinded, randomized clinical trial	Treatment	Industry	United States, China, Denmark, India, Malaysia, Mexico, Singapore, South Africa, Taiwan, Thailand, and Vietnam	No
NCT05335876	Not yet recruiting	Onasemnogene abeparvovec	Child, Adult, Older Adult	3	260	Interventional	Single-group, open-label clinical trial	Treatment	Industry	N.R.	No
NCT05386680	Not yet recruiting	Onasemnogene abeparvovec	Child	3	28	Interventional	Single-group, open-label clinical trial	Treatment	Industry	N.R.	No
NCT03955679	Unknown status	Onasemnogene abeparvovec	Child, Adult, Older Adult	N.A.	N.A.	Expanded access	N.R.	N.R.	Industry	United States	No
NCT03421977	Active, not recruiting	Onasemnogene abeparvovec	Child, Adult, Older Adult	N.A.	13	Observational	Prospective observational study	N.R.	Industry	United States	Yes
NCT04174157	Recruiting	Onasemnogene abeparvovec	Child, Adult, Older Adult	N.A.	500	Observational	Cross-sectional study	N.R.	Industry	United States	Yes
NCT03779334	Active, not recruiting	Risdiplam	Child	2	25	Interventional	Single-group, open-label clinical trial	Treatment	Industry	United States, Australia, Brazil, China, Poland, Russia, and Taiwan	Yes
NCT03920865	Completed	Risdiplam	Adult, Older Adult	1	26	Interventional	Non-randomized, open-label clinical trial	Treatment	Industry	United States	No
NCT03988907	Completed	Risdiplam	Adult	1	35	Interventional	Non-randomized, open-label clinical trial	Treatment	Industry	United States	No
NCT04718181	Recruiting	Risdiplam	Adult	1	268	Interventional	Open-label, randomized clinical trial	N.R.	Industry	United States	No
NCT05522361	Not yet recruiting	Risdiplam	Child, Adult	4	10	Interventional	Single-group, open-label clinical trial	Treatment	Industry	United States	No
NCT05115110	Recruiting	Risdiplam	Child	2/3	180	Interventional	Double-blinded, randomized clinical trial	Treatment	Industry	Belgium, Italy, The Netherlands, Poland, and United Kingdom	No
NCT05232929	Recruiting	Risdiplam	Child, Adult, Older Adult	4	500	Interventional	Single-group, open-label clinical trial	Treatment	Industry	United States	No
NCT02633709	Completed	Risdiplam	Adult	1	33	Interventional	Double-blinded, randomized clinical trial	Treatment	Industry	The Netherlands	No
NCT02908685	Active, not recruiting	Risdiplam	Child, Adult	2/3	231	Interventional	Double-blinded, randomized clinical trial	Treatment	Industry	United States, Belgium, Brazil, Canada, China, Croatia, France, Germany, Italy, and Japan	Yes
NCT02913482	Active, not recruiting	Risdiplam	Child	2/3	62	Interventional	Non-randomized, open-label clinical trial	Treatment	Industry	United States, Belgium, Brazil, China, Croatia, France, Italy, Japan Poland, Saudi Arabia, Spain, and Switzerland	Yes
NCT03032172	Active, not recruiting	Risdiplam	Child, Adult	2	174	Interventional	Single-group, open-label clinical trial	Treatment	Industry	United States, Belgium, France, Germany, Italy, The Netherlands, Switzerland, and United Kingdom	Yes
NCT04256265	Unknown status	Risdiplam	Child, Adult, Older Adult	N.A.	N.A.	Expanded access	N.R.	N.R.	Industry	United States	No
NCT05219487	Recruiting	Risdiplam	Adult, Older Adult	N.A.	10	Observational	Prospective observational study	N.R.	Industry	United States	No
NCT04177134	Recruiting	Nusinersen, onasemnogene abeparvovec, risdiplam	Child, Adult, Older Adult	N.A.	1000	Observational	Cohort study	N.R.	University	France	No
NCT05518773	Recruiting	Nusinersen, risdiplam	Child, Adult	N.A.	34	Observational	Cross-sectional study	N.R.	Industry	United States	No
NCT03339830	Completed	N.R.	Child, Adult, Older Adult	N.A.	100	Observational	Prospective cohort study	N.R.	Research center	France	No
NCT05475691	Recruiting	N.R.	Child, Adult, Older Adult	N.A.	300	Observational	Prospective cohort study	N.R.	Industry	Argentina, Brazil, Mexico, and Uruguay	No
NCT05618379	Not yet recruiting	N.R.	Adult, Older Adult	N.A.	200	Observational	Prospective observational study	N.R.	Industry	N.R.	No
EU PAS register											
EUPAS32033	Planned	Nusinersen	Child, Adult, Older Adult	N.A.	300	Observational	Intensive monitoring scheme	Biomarker discovery	Industry	Germany	No
EUPAS41853	Ongoing	Onasemnogene abeparvovec	Child, Adult	N.A.	500	Observational	Observational registry	Effectiveness evaluation	Industry	Argentina, Brazil, Chile, Greece, Ireland, Japan, Republic of Korea, Poland, Portugal, Romania, Russia, Taiwan, and United States	No
EUPAS47916	Ongoing	Risdiplam	Child, Adult, Older Adult	N.A.	600	Observational	Cohort study	Effectiveness evaluation	Industry	Austria, Germany, Sweden, and Switzerland	No
EUPAS47679	Planned	Risdiplam	Adolescent, Adult	N.A.	46	Observational	Cohort study	Safety evaluation	Industry	Germany, Italy, and United States	Yes

Abbreviations: N.A. = Not applicable; N.R. = Not reported.

## Data Availability

Not applicable.

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
