# Peer review of "Pharmacological Therapies of Spinal Muscular Atrophy: A Narrative Review of Preclinical, Clinical–Experimental, and Real-World Evidence"

_brainsci, 2023, doi:10.3390/brainsci13101446_

Round 1
Reviewer 1 Report
This is a very-well written piece cataloguing the studies supporting the use of three SMA medicines, including post-approval studies.
Figure 1 is a nice summary of the points of potential intervention.
Abstract and title: the title implies that the paper will describe the benefit offered by the three drugs. Unexpectedly, the abstract does not provide this information. Tell readers in the abstract what you found; or, rephrase the title to shift the focus from “evidence” to a “catalogue of studies” or something similar.
In general, many points in the article could benefit from greater precision. For example:
- “90% of cases” is vague – Are the authors referring to 90% of patients? 90% of SMN2 transcripts? Rephrase to be more precise.
- “solid path” is vague. Do the authors mean: faster approval ? Approval based on more limited data? What “strategies”? Who are the “stakeholders”? What is different about the PK/PD profile, and why do those differences matter? Greater specificity could help readers understand what you are arguing.
- “improved lifespan” à be more specific (e.g., state the number of days/weeks with the drug to number of days/weeks without the drug). Same for “significant recovery of motor function”
- “significantly improved lifespan” ; same comment as above (state the number of days/weeks…)
- “compared to 32% [13/41] of patients in the control group” - over what period of time? How long did the 61% of patient live?
- “confirmed the safety and the efficacy over time” – what was the efficacy? Can it be quantified in easy-to-understand terms? E.g., Of X patients, how many were still alive at age Y?
- “a stabilization of disease progression” à vague; can their ages and motor abilities be described with greater specificity?
- “maintenance of the motor mile-110 stones previously achieved, even after 7.5 years” à good to specify 7.5 years, but what motor milestones were achieved? How did development compare to healthy children at the same age? What normal activities were they still unable to perform? What activities could they perform just fine?
- “two fatal cases” … out of how many? without a denominator, the risk cannot be easily understood by patients or providers.
- “lack of efficacy in controlling peripheral symptoms” …vague; what “peripheral symptoms” were there?
- “MFM32 score was 1.36 “ -- tell readers what the range of the scale is (0 to 32?); without a scale, the 1.36 figure is meaningless.
- “a treatment difference of 1.55” – same comment. If the scale is out of 32, then this is a 4.8% difference. Do parents realize this is the amount of benefit they can expect?
Overall, I would encourage the authors to shift the focus of the article to clarifying the amount of benefit that patients can expect to receive from the drug. How long will they live? What level of motor development can they expect to have, and for how long? What can they do? What can’t they do? Will they survive until age 8, but need a ventilator and feeding tube for the last 6 years of life? These are the types of questions patients care about, but the way the evidence is presented in this paper, it is not possible for patients (or their caregivers) to know the answers.
A few minor errors:
- “requires to make the study protocols “ – requires *who* to make the study protocols? Researchers? A word is missing.
- “patients-derived” should be “patient-derived”; “compounds optimization” should be “compound optimization” “posed” should be “provided”
- “share a similar nusinersen’s mechanism of action” – rephrase; a word is missing
see above
Author Response
This is a very-well written piece cataloguing the studies supporting the use of three SMA medicines, including post-approval studies.
Figure 1 is a nice summary of the points of potential intervention.
Comment: Abstract and title: the title implies that the paper will describe the benefit offered by the three drugs. Unexpectedly, the abstract does not provide this information. Tell readers in the abstract what you found; or, rephrase the title to shift the focus from “evidence” to a “catalogue of studies” or something similar.
Response: We thank the Reviewer for this comment. We added the following phrase in the abstract: “As of January 4th, 2023, a total of 69 post-marketing studies concerning the three orphan drugs approved for SMA were identified in clinicaltrials.gov (N= 65; 94.2%) and EU PAS Register (N= 4; 5.8%). Currently ongoing studies are primarily aimed to provide evidence concerning the risk/benefit profile of the three drugs in specific populations which were not included in the pivotal trials and to investigate the long-term safety and clinical benefits of these drugs.”
Comment: In general, many points in the article could benefit from greater precision. For example:
- “90% of cases” is vague – Are the authors referring to 90% of patients? 90% of SMN2 transcripts? Rephrase to be more precise.
Response: We thank the Reviewer for this comment. We rephrased as follows: “In 90% of SMN2 transcriptions, the SMN2 transcript undergoes splicing of exon 7, thus yielding an unstable SMN∆7 protein, while in the remaining 10% a normal and functional SMN protein is produced, contributing to the survival of the spinal motor neurons and influencing the severity of the clinical manifestations.”
- “solid path” is vague. Do the authors mean: faster approval? Approval based on more limited data? What “strategies”? Who are the “stakeholders”? What is different about the PK/PD profile, and why do those differences matter? Greater specificity could help readers understand what you are arguing.
Response: We thank the Reviewer for this comment. With this paragraph, we meant to say that the efforts of different stakeholders (i.e., research centers, industries, and regulators) are crucial to gain robust evidence supporting regulatory processes and to obtain innovative therapies. We edited the text accordingly.
- “improved lifespan” à be more specific (e.g., state the number of days/weeks with the drug to number of days/weeks without the drug). Same for “significant recovery of motor function”
Response: We thank the Reviewer for this comment. We edited the text as follows: “Treatments markedly increased SMN protein levels and ameliorated the disease phenotype, with an extended median lifespan (severely reduced in SMA mice) by 25-fold, and a significant recovery of motor function (including α-motor neuron count, mean area of muscle fibers, heart weight, the thickness intraventricular septum, the left ventricular wall, and the integrity of neuromuscular junctions), that was relevant from a clinical perspective [41].”
- “significantly improved lifespan”; same comment as above (state the number of days/weeks…)
Response: We thank the Reviewer for the comment. We added to the text more detailed information regarding the improved lifespan of treated mice with the relative reference. We edited the text as follows: “(median of 282 days for treated mice vs. 17.5 days for untreated SMAΔ7 mice)”.
- “compared to 32% [13/41] of patients in the control group” - over what period of time? How long did the 61% of patient live?
Response: We thank the Reviewer for this comment. We edited as follows: “61% (49/80) of patients did not die or did not require permanent ventilation (compared to 32% [13/41] of patients in the control group; p=0.005) at the end-of-trial visit (i.e., day 183, 302, or 394)”. The information regarding the lifespan of the 61% of patients was not reported in the trial.
- “confirmed the safety and the efficacy over time” – what was the efficacy? Can it be quantified in easy-to-understand terms? g., Of X patients, how many were still alive at age Y?
Response: We thank the Reviewer for this comment. We added the following sentence: “The study confirmed the safety and the efficacy over time of the therapy, with evidence of a mild but significant amelioration of disease-related neuromuscular symptoms evaluated through the Children's Hospital of Philadelphia Infant Test of Neuromuscular Disorders (CHOP INTEND) and the Hammersmith Infant Neurological Examination (HINE-II). During the 4-year follow-up period, only one patient receiving nusinersen died between the second and the fourth year of follow-up.”
- “a stabilization of disease progression” à vague; can their ages and motor abilities be described with greater specificity?
Response: We thank the Reviewer for this comment. We edited the text as follows: “A prospective 3-year registry study (SMArtCARE), published in January 2023, highlighted a stabilization of disease progression in 231 ambulant patients (114 pediatric patients with a median age of 8.6 years and 117 adult patients with a median age of 37 years), in parallel with an improvement of motor function [31]. In total, 31 pediatric patients (27.2%) and 31 adult patients (26.5%) achieved an improvement of more than 30 meters in the 6-Minute-WalkTest. On the contrary, among the patients receiving nusinersen, only five adult patients (7.7%) experienced a decline in walking distance of more than 30 meters, and two pediatric patients (1.8%) lost the ability to walk without assistance.”
- “maintenance of the motor mile-110 stones previously achieved, even after 7.5 years” à good to specify 7.5 years, but what motor milestones were achieved? How did development compare to healthy children at the same age? What normal activities were they still unable to perform? What activities could they perform just fine?
Response: We thank the Reviewer for this comment. The LT-002 study on onasemnogene abeparvovec is still ongoing and, unfortunately, we do not have this information. We only added that the maintenance of a motor milestone was assessed according to the Developmental Milestone Checklist.
- “two fatal cases” … out of how many? without a denominator, the risk cannot be easily understood by patients or providers.
Response: We thank the Reviewer for this comment. This data does not come from clinical studies, but from spontaneous reporting systems. As such, so it is not possible to have a denominator.
- “lack of efficacy in controlling peripheral symptoms” …vague; what “peripheral symptoms” were there?
Response: We thank the Reviewer for this comment. We have better detailed the meaning of peripheral symptoms and added the relative references. We edited the text as follows: “In fact, the SMN protein is also located in extra central nervous system organs (such as muscle, heart, pancreas, afferent nerves, bones, gastroenteric system) and low levels of SMN induce a wide variety of non-motor symptoms (e.g., impaired skeletal muscle development, cardiomyopathy, altered liver function, severe diarrhea, and lower bone density).”
- “MFM32 score was 1.36 “ - tell readers what the range of the scale is (0 to 32?); without a scale, the 1.36 figure is meaningless.
Response: We thank the Reviewer for this comment. The authors of the paper did not report the mean value of the MFM32 scale for the intervention and the control group, but they only report the mean change from baseline (1.36 vs -1.22, respectively). Absolute MFM32 scores range from 0 to 96 (i.e., 32 items, each with a maximum score of 3), with higher scores indicating better motor function.
- “a treatment difference of 1.55” – same comment. If the scale is out of 32, then this is a 4.8% difference. Do parents realize this is the amount of benefit they can expect?
Response: We thank the Reviewer for this comment. The authors of the paper did not report the mean value of the MFM32 scale for the intervention and the control group, but they only report the mean change from baseline (1.36 vs -1.22, respectively). Absolute MFM32 scores range from 0 to 96 (i.e., 32 items, each with a maximum score of 3), with higher scores indicating better motor function.
Comment: Overall, I would encourage the authors to shift the focus of the article to clarifying the amount of benefit that patients can expect to receive from the drug. How long will they live? What level of motor development can they expect to have, and for how long? What can they do? What can’t they do? Will they survive until age 8, but need a ventilator and feeding tube for the last 6 years of life? These are the types of questions patients care about, but the way the evidence is presented in this paper, it is not possible for patients (or their caregivers) to know the answers.
Response: We agree with the reviewer about the importance of the questions raised, especially for the patients and caregivers. However, caution is necessary to avoid excessive expectations, since longer follow-up studies are needed to have a clearer view about the extent and the duration of the benefits observed, as well as the long-term safety. This is a very dynamic field and further improvement in patients care is expected with the available therapies and those under development.
A few minor errors:
- “requires to make the study protocols “ – requires *who* to make the study protocols? Researchers? A word is missing.
Response: We thank the Reviewer for this comment. We edited the text as follows: “The current European pharmacovigilance legislation requires study protocols and summary of results to be publicly available for PASS imposed as an obligation of marketing authorization by competent authorities.”
- “patients-derived” should be “patient-derived”; “compounds optimization” should be “compound optimization” “posed” should be “provided”
Response: We thank the Reviewer for this comment. We edited the text as suggested.
- “share a similar nusinersen’s mechanism of action” – rephrase; a word is missing
Response: We thank the Reviewer for this comment. We edited the text as follows: “Innovative drug design techniques in combination with high-throughput screening methods permitted the identification of novel small molecules that sharing nusinersen’s mechanism of action, promoting the inclusion of exon 7.”

Reviewer 2 Report
The authors provided a nice and well-written narrative review about 5q SMA therapies. This is a complex clinical scenario currently present in the practice of both neuromuscular diseases specialists and neuropediatricians. This manuscript will certainly summarize the main data related to preclinical, clinical and real-world data about the three main therapies used currently in the treatment of patients with 5q SMA: nusinersen, risdiplam, and onasemnogene abeparvovec. The manuscript structure shows all information in an easy and accessible way, summarizing complex clinical trials which occurred mainly in the last five years. Here are some suggestions to even improve the high quality of the manuscript:
1. Authors should take a look at the presentation of the gene names in the manuscript, which must be done with italics - SMN1, SMN2 (line 33, line 40, line 46, line 81, line 82, line 85, Table 1; lines 5, 6, 30 and 31 in the topic 2.1.1. and 2.1.2. - line 13 in the same topic has a minor typo "at a time4"). The name of Caenorhabditis elegans must also be present in italics (line 78).
2. I suggest authors to add their references in Table 2 to make it easier for the reader to look for the original texts.
3. In Table 2, in the cell describing the eligibility criteria for the STR1VE-US study, it is described "(...) than 6 months of age at the date of Risdiplam administration". Authors should review this criteria.
4. There is a minor typo in the Conclusion to be corrected: "5. . Conclusion".
Author Response
The authors provided a nice and well-written narrative review about 5q SMA therapies. This is a complex clinical scenario currently present in the practice of both neuromuscular diseases specialists and neuropediatricians. This manuscript will certainly summarize the main data related to preclinical, clinical and real-world data about the three main therapies used currently in the treatment of patients with 5q SMA: nusinersen, risdiplam, and onasemnogene abeparvovec. The manuscript structure shows all information in an easy and accessible way, summarizing complex clinical trials which occurred mainly in the last five years. Here are some suggestions to even improve the high quality of the manuscript:
Comment: 1. Authors should take a look at the presentation of the gene names in the manuscript, which must be done with italics - SMN1, SMN2 (line 33, line 40, line 46, line 81, line 82, line 85, Table 1; lines 5, 6, 30 and 31 in the topic 2.1.1. and 2.1.2. - line 13 in the same topic has a minor typo "at a time4"). The name of Caenorhabditis elegans must also be present in italics (line 78).
Response: We thank the Reviewer for this comment. We edited the text as suggested.
Comment: 2. I suggest authors to add their references in Table 2 to make it easier for the reader to look for the original texts.
Response: We thank the Reviewer for this comment. We added references in Table 2.
Comment: 3. In Table 2, in the cell describing the eligibility criteria for the STR1VE-US study, it is described "(...) than 6 months of age at the date of Risdiplam administration". Authors should review this criteria.
Response: We thank the Reviewer for this comment. We addressed this typo in the table.
Comment: 4. There is a minor typo in the Conclusion to be corrected: "5. . Conclusion".
Response: We thank the Reviewer for this comment. We edited the text as suggested.
